# SAR Target Recognition with Limited Training Samples in Open Set Conditions

**DOI:** 10.3390/s23031668

**Published:** 2023-02-02

**Authors:** Xiangyu Zhou, Yifan Zhang, Di Liu, Qianru Wei

**Affiliations:** School of Software, Northwestern Polytechnical University, Xi’an 710129, China

**Keywords:** synthetic aperture radar (SAR), limited training samples, open set recognition (OSR), graph convolutional network (GNN), relation measure

## Abstract

It is difficult to collect training samples for all types of synthetic aperture radar (SAR) targets. A realistic problem comes when unseen categories exist that are not included in training and benchmark data at the time of recognition, which is defined as open set recognition (OSR). Without the aid of side-information, generalized OSR methods used on ordinary optical images are usually not suitable for SAR images. In addition, OSR methods that require a large number of samples to participate in training are also not suitable for SAR images with the realistic situation of collection difficulty. In this regard, a task-oriented OSR method for SAR is proposed by distribution construction and relation measures to recognize targets of seen and unseen categories with limited training samples, and without any other simulation information. The method can judge category similarity to explain the unseen category. Distribution construction is realized by the graph convolutional network. The experimental results on the MSTAR dataset show that this method has a good recognition effect for the targets of both seen and unseen categories and excellent interpretation ability for unseen targets. Specifically, while recognition accuracy for seen targets remains above 95%, the recognition accuracy for unseen targets reaches 67% for the three-type classification problem, and 53% for the five-type classification problem.

## 1. Introduction

Most of the existing synthetic aperture radar (SAR) target recognition methods are aimed at closed set conditions, in which the categories of targets in the test set have appeared in the training set. However, the collection of SAR images is difficult. In practical conditions, the existing training set only contains a limited number of categories, and the classifier is likely to encounter targets from unseen categories. This practical condition is called the open set condition [1].

To handle this challenge, open set recognition (OSR) methods have been developed. OSR usually requires the classifiers to not only accurately classify seen categories, but to also effectively deal with unseen categories. Unlike generalized OSR, which has side-information (e.g., semantic/attribute information, etc.) involved in the training, the unseen categories in OSR for SAR targets usually refer to unknown unknown classes (UUCs), i.e., classes that have no information regarding them during training: not only unseen but also having no side-information during training [2,3].

Some scholars have shed light on the OSR methods for SAR. Most of the existing methods construct a reference space and identify the target unseen category according to the location of it in the space. The recognition effect depends on the quality of the reference space. Taken into account, these methods usually require a large number of samples for training to construct a high-quality reference space. However, due to the difficulty of SAR image collection and marking, it is often impossible to obtain enough samples. In this case, these methods cannot achieve the desired effect.

To work well under the conditions of limited training samples, the idea of task-oriented training in few shot learning (FSL) [4,5,6,7,8,9,10] can be combined. However, not all FSL methods are applicable. Only the methods based on learning distribution are applicable to OSR, while the methods based on learning features are completely ineffective when it comes to unseen categories. The graph convolutional network (GNN) is inherently suitable for learning distribution. GNN has been widely used in the field of target recognition [11], and it provides new ideas for the OSR of SAR targets. GNN can exploit the relationships between the features of samples and is beneficial for structural relationship representations [12,13].

Therefore, this paper proposes an OSR method for SAR based on GNN, which is used to build a network of relationships between various types of images. The method classifies targets of seen categories while rejecting targets of unseen categories through the density of relationships, and interprets the unseen category through the correlation between this category and other categories. The training process is task-oriented to acquire the ability to quickly construct distribution when encountering a new task containing a few samples for support. The uniqueness of the proposed method is that it does not require side-information in training and can obtain good recognition results when the number of training samples is limited. In summary, our contributions lie in the following aspects:We are the first to explore the problem of SAR OSR with limited training samples, which has practical implications.We introduce the GNN to SAR OSR; the proposed method can identify targets from seen and unseen categories well.For the target of the unseen category, the proposed method can not only identify but also interpret it by distance measurement. The method has stability for different recognition tasks.

## 2. Related Work

### 2.1. Generalized Open Set Recognition

The definition of generalized OSR is that when a model trained on the training set with side-information is tested with a test set, which contains unseen categories, if the sample of seen categories are input, specific categories will be output; if the sample of unseen categories are input, appropriate processing will be carried out (identified as unseen). Since Scheirer et al. introduced the concept of OSR in [1], lots of generalized OSR methods have been successful in the fields of computer vision and optical image classification.

For example, Bendate et al. [14] solved OSR by introducing a new model layer named OpenMax, which estimates the probability of an input being from an unseen category. Scherreik and Rigling [2] proposed a new method, named the probabilistic open set support vector machine (POS-SVM), for OSR. In addition, [2] provided the first detailed and explicit description of OSR. Bapst et al. [15] designed an aerial image ATR classifier by using the synthesized images of unseen categories of targets. Rudd et al. [16] established an extreme value machine (EVM) for OSR. EVM is well explained from statistical extreme value theory (EVT), and it is the first classifier that can perform nonlinear non-core variable bandwidth incremental learning. Dang et al. [17] improved the open set model with the incremental learning (OSmIL) method with incremental learning, which can continually recognize and learn new categories.

### 2.2. OSR for SAR Target

Different from generalized OSR, for SAR OSR, there is a lack of information assistance in training because semantic annotation of SAR images is difficult. Some scholars have studied OSR algorithms for SAR targets. Scherreik and Rigling [18] improved the POS-SVM method proposed in [2] to design a Weibull-calibrated SVM (W-SVM) and applied it to the OSR of SAR targets. Toizumi et al. [19] introduced optical images to construct a high-quality reference space to recognize unseen categories. Song and Xu [20] used a deep generative neural network (DGNN) to construct a continuous SAR target reference space for OSR. To improve the clustering degree of objects in the reference space, Song et al. [21] added some simulated images as auxiliary data during the construction process of the reference space. Dang et al. [22] proposed an edge exemplar selection method, named O_SAR, to extract class boundaries, based on the local statistical information. Both Song et al. [21] and Dang et al. [22] methods work well in OSR; however, they need a lot of prior knowledge of unseen categories, which limits their applicability. Wei et al. [23] improved the quality of the reference space without prior knowledge by introducing the idea of the adversarial network. Ma et al. [24] proposed an OSR method based on multitask learning, which was also developed from an adversarial network. Zeng et al. [25] proposed the Fea-DA to finely identify the seen and unseen targets by calculating the relative position angle, which achieved the state-of-the-art (SOTA). Although these methods do not require prior knowledge, they require a large number of samples for training, which is not easy to achieve in reality either.

### 2.3. GNN-Based Methods in FSL

In recent years, GNN has been widely used as an important method in the area of few-shot learning. In particular, Garcia et al. [12] used GNN to resolve the FSL problem first, where the embedding model and GNN model were trained end-to-end as one. Liu et al. [26] put forward a transductive propagation network (TPN). The TPN uses the whole query set for transductive inference to further exploit intra-class similarity and inter-class dissimilarity. Gidaris et al. [27] used a denoising autoencoder network to reconstruct the classification weights of the GNN-based few-shot model. Kim et al. [28] put forward an edge-labeling graph neural network. To model the relationship of distribution level explicitly, Yang et al. [29] put forward a distribution propagation graph network (DPGN). GNN-based models have interpretability and good performance, which deserve extensive research.

## 3. Materials and Methods

An OSR with limited training samples task is defined as a (C + 1)-types classification problem, with C seen categories and an unseen category, and there are K samples for each seen category as support. The training of the model is task-oriented, combining the idea of meta-learning [30]. We set multiple groups of recognition tasks and conduct supervised training for each group of tasks.

As shown in Figure 1, the basic features of different categories of samples are extracted by a feature extractor and sent into the relation space for distance measurement. A fully-connected (FC) layer is connected at the end to output the recognition result.

### 3.1. Feature Extraction

The feature extractor $\varphi \left(\cdot \right)$ is used for basic feature extraction, i.e., processing SAR images into one-dimensional vectors of length 64. As shown in Figure 2, the feature extractor includes a group of convolutional layers to map the high-dimensional vector to the low-dimensional vector. Several attention modules are interspersed among them to learn the correlation between channels and change the weight of each channel [31]. Through these modules, important information can be enhanced to make the extracted features more directional.

The attention module firstly conducts a global average pooling of the feature map M ∈ ℝH×W×C obtained by convolution:(1)Z=1H×W∑i=1H∑i=1Wmci,j
where Z ∈ ℝC is the channel-wise statistic mc∈M =m1,m2,···,mC. Then, an FC layer is connected to learn the dependencies between channels. Finally, the Sigmoid activation function is used to limit the value to the range of 0,1 and obtain a one-dimensional vector E as the evaluation score:(2)E=σραZ
where σ· represents the Sigmoid activation function, ρ· represents the ReLU activation function, and α is a learned weight parameter. M multiplied by E as the feature map of the next convolution layer:(3)M*=EM

After obtaining the basic feature $\varphi \left(x_{i}\right)$ of all samples, the label information is concatenated with the basic feature as the original feature $F_{i}$ input into the relation space:(4)$$F_{i}=\left(\varphi \left(x_{i}\right),h\left(y_{i}\right)\right)$$
where $y_{i}$ is the true label of sample $x_{i}$, and $h\left(y_{i}\right)$ represents the one-hot coding of $y_{i}$.

### 3.2. Relationship Measurement

Relationships are measured in the relation space, which is essentially a fully connected graph neural network. There are C × K +1 nodes, which are the features of all samples in a task T input into the graph network. The graph is defined as:(5)$$G_{T}=\left(V=\left\{v_{i}\right\},R=\left\{r_{i,j}\right\}\right)$$
where $r_{i,j}$ is the edge between node $v_{i}$ and node $v_{j}$, representing the similarity between $v_{i}$ and $v_{j}$. $v_{i}=F_{i}$ at the beginning.

As shown in Figure 3, the relation space consists of three adjacency modules and two update modules alternately stacked. After the initial nodes are input into the relation space, the following operations are repeated:Construct the adjacency matrix (adjacency module): The relationship between every two nodes is expressed as:
(6)$$r_{i,j}=\mathrm{MLP}\left(\mathrm{abs}\left(v_{i},v_{j}\right)\right)$$

where MLP is a multi-layer neural network, for which the input is the Euclidean distance between $v_{i}$ and $v_{j}$. After obtaining all $r_{i,j}$, adjacency matrix R can be constructed.

Change the features (update module and concatenation): The new features are learned by the following equation:


(7)
$$V_{l-1}{}^{\prime }=\rho \left({\beta \mathrm{RV}}_{l-1}\right)$$


where ρ represents the update module (contains FC and ReLU layers, the former for tensor deformation and the latter for the nonlinear activation function). β is a learned weight parameter. $V_{l-1}$ is the matrix of features before the lth iteration, and $V_{0}=\left\{F_{i}\right\}$. Concatenate previous features with learned features as: (8)$$V_{l}=\left(V_{l-1},V_{l-1}{}^{\prime }\right)$$

### 3.3. Strategies in Training and Testing Phase

For a dataset, we randomly divide all samples of one category into the unseen set $D_{u}$ and the samples in the remaining categories into seen set $D_{s}$. For each category in $D_{s}$, we randomly select 80% of the samples to form $D_{\mathrm{st}}$ for training, and the remaining 20% of the samples to form $D_{\mathrm{se}}$ for testing. $D_{u}$ contains the true unseen categories. The samples in $D_{\mathrm{se}}$ and $D_{u}$ together only participate in testing, not in training. The samples in $D_{\mathrm{st}}$ only participate in training, and some of the categories act as simulated unseen categories.

During a training task, as shown in Table 1, K samples of C categories from Dst are selected to construct support set Ts, and a sample of those C categories (as seen category) or one of other categories from Dst (as unseen category) is selected to construct query set Tq; Ts ∩ Tq= ø. All samples in Ts and Tq are input into the model, and the label of Tq sample is used as the ground-truth for prediction, to form supervised training. The cross-entropy loss of the possibility distribution of categories output by the FC layer is calculated as follows:
(9)L=∑i=1C+1yquelogy^que
where yque is the label of query sample xque in Tq, and y^que is output by the model and represents the confidence The parameters of the whole model are updated by back-propagation as follows: (10)θ′=θ−η∇Lθ
where θ is the previous parameter set, ∇Lθ represents the partial derivative of the loss with respect to θ, and η is the learning rate. The training details are shown in Algorithm 1 (see Appendix A).
**Algorithm 1:** Training phase.**Input:** Learning rate η, the whole model g, task distribution pT, the number of batches M, the batch size I**Output:** good parameters θ*1: Randomly initialize θ;2: **for** m=1; m≤M; m=m+1 **do**3:   Sample I tasks from pT;4:   **for** j=1; j≤I; j=j+1 **do**5:     Get initial features F=φxi,hyii∈1,C×K+1;6:     Get the matrix of initial features V0;7:     **for** k=1; k≤2; k=k+1 **do**8:         ri,j=MLPabsvi,vj;9:         Get the adjacency matrix Rk−1;10:          V’k=ρβk−1Rk−1Vk−1;11:          Vk=Vk−1,V’k;12:     **end for**13:     Get the final adjacency matrix Rfinal;14:     Get the categories possibility distribution y^que;15:     L=∑i=1C+1yquelogy^que;16:   **end for**17:   Backward ∑j=1IL as ∇L;18:   Update θ′=θ−η∇Lθ;19: **end for**

During a testing task, as shown in Table 1, K samples of C categories from $D_{\mathrm{se}}$ are selected to construct $T_{s}$, and a sample from $D_{u}$ or the C categories is selected to construct query set $T_{q}$ as the target to be recognized. All samples in $T_{s}$ and $T_{q}$ are input into the model with fixed parameters to obtain the possibility distribution of categories, which contains recognition results and similarity information. If the highest probability is category C +1, the prediction label is “unseen”, and the unseen category is similar to the high probability category. If not, the prediction label is the highest probability category.

## 4. Experiment and Discussion

We evaluate our method on the MSTAR dataset and sample under the standard operating condition (SOC) with two depression angles (15° and 17°) [32], including 10 types of ground targets (T62, BTR60, ZSU234, BMP2, ZIL131, T72, BTR70, 2S1, BRDM2, and D7). Optical images of various targets and their corresponding SAR images are shown in Figure 4. T72 is set as the unseen set $D_{u}$ unless otherwise specified. The objects of the same category with different depression angles are mixed. All images are processed to size 100 × 100.

All of the experiments were implemented in Python 3.7, on a workstation with the Intel Core i7-11700K CPU (eight cores, 3.80 GHz) and the NVIDIA GeForce RTX 3060 GPU (12-GB memory). The Adam optimizer with a learning rate of 0.001 was adopted. To balance the performance of seen and unseen categories recognition, we used the seen category and unseen categories as $T_{q}$ in a ratio of 1:1 throughout the training process.

To prove the effectiveness of the proposed method, we compared it with several existing OSR methods (Fea-DA [19], GAN_OSR [18], and O_SAR [16]), and a classical FSL method, the prototype network [3]. The recognition results of the five-types classification problem (K = 20) are shown in Table 2. It can be seen that the recognition accuracies of Fea-DA and GAN_OSR are worse than that of the proposed method, while the prototype network simply does not work. Although O_SAR performs better for unseen categories, it is particularly poor for seen categories. The analyses are as follows: (1) Faced with the situation that different unseen categories participate in the training, the prototype network, a representation-learning-based method, cannot generate stable feature mapping, while the distribution generated by GNN is relatively stable. (2) Faced with the situation of limited training samples, the quality of the reference space obtained by Fea-DA and GAN_OSR, which are both based on the generative network, is too low to achieve good results. (3) O_SAR is a class boundary extraction method based on local statistical information. It is easily affected by the chance of sampling, and its recognition effect is extremely unstable when the sampling amount is too small.

The distribution of features of various seen categories through the prototype network and the proposed method in a five-types classification training task (K = 20) are visualized by t-SNE [33] in Figure 5. The results show that the prototype network cannot effectively cluster in OSR, while the proposed model can cluster the samples of the same category together, and the different categories are separated.

The effectiveness of the proposed method was proved from the two aspects of recognition accuracy and interpretability.

(1) Recognition Accuracy: For each case of different settings, we set 5000 testing tasks to obtain the average accuracy. As shown in Table 3, for three-, five-, and seven-types classification problems, the recognition accuracy of seen categories can reach 99.06%, 96.48%, and 94.27%, respectively; the recognition accuracy of unseen categories can reach 67.00%, 52.76%, and 34.12%, respectively. There are C +1 types of output for both “seen categories” and “unseen categories”. Therefore, if the outputs are random, the accuracies of both of them are 25%, 16.7%, and 12.8% for three, five, and seven classification problems. The above proves the proposed method has a good ability to identify.

(2) Interpretability: As shown in Figure 6a,b, in a testing task, T62 (tank), D7 (bulldozer), BMP2 (infantry fighting vehicle), BRDM2 (armored reconnaissance vehicle), and 2S1 (self-propelled howitzer) are selected as the $T_{s}$. For Figure 6a,b, BMP2 and T72 (tank) are respectively taken as the $T_{q}$. It can be seen that BMP2 is well identified, while T72 is well separated from the other categories (i.e., identified as unseen). By distance comparison, T72 and T62 are visually similar. As shown in Figure 6c,d, in another testing task, five categories of D7, BRDM2, BMP2, BTR70 (armored transport vehicle), and ZIL131 (freight truck) are selected as $T_{s}$, T72 and BRDM2 are used as $T_{q}$ for Figure 6c,d, respectively. It can be seen that BRDM2 is well identified, and T72 is well separated from the other categories, as before. From the distance comparison, it can be seen that T72 is similar to BMP2 and BTR70 in these five categories, which is also in line with reality. It can be seen from Figure 6e,f that different samples of the same category as input will not cause significant changes in the distance between categories. The above can prove the proposed method has an excellent and stable ability to interpret.

Each of the ten categories was successively treated as the unseen set $D_{u}$ for training, and then one hundred rounds of testing were conducted, in which five categories were randomly selected from the remaining nine categories as support categories for each round of testing. The frequencies of each predicted category were calculated as a heat map. We constructed heat maps for the proposed method, Fea-DA, and GAN_OSR, as above. As shown in Figure 7a, most of the predicted results of the proposed method are correct. In addition, it can be seen that the non-zero elements in the matrix are relatively symmetric along the diagonal; i.e., when the target is not identified as an unseen category, it is identified as a relatively similar category. This indicates that the distribution distance of similar categories through the proposed model is relatively close. Hence, the model has the ability to explain the location category through the distance metric. In contrast, the non-zero elements in Figure 7b are not symmetric along the diagonal, and, in Figure 7c, a significant proportion of the predictions are incorrect.

A study on the setting of the ratio of the seen category and unseen category as $T_{q}$ during training was carried out. That is, the study aimed to explore the influence of focusing training on matching seen targets or rejecting unseen targets on the recognition effect. The results in three- and five-types classification tasks (K = 20) are shown in Table 4. It can be seen that properly increasing the proportion of the unseen category as $T_{q}$ during training can greatly improve the accuracy of unseen category recognition without having much impact on the recognition effect of all categories.

## 5. Conclusions

In this paper, we proposed an effective method for SAR OSR with limited training samples and without any other simulation information, different from the existing methods and challenging. In this method, GNN was used to construct a space of relationships among various types of images. The seen and unseen categories were identified according to the density of the relationship. Experimental results on the MSTAR dataset showed that the proposed method could achieve the state-of-the-art compared with the existing OSR methods under the same restrictions. Specifically, the experimental results of three-type classification problem results are: a seen target accuracy of 99.06%, an unseen target accuracy of 67.00%; the five-type classification problem results are: a seen target accuracy of 96.00%, an unseen target accuracy of 52.76%; and the seven-type classification problem results are: a seen target accuracy of 94.27%, and an unseen target accuracy of 34.12%. In addition, this method not only has good recognition accuracy for unseen targets, but also has the ability to interpret them, which is not available in other methods. However, the performance of the proposed OSR method will decrease when the number of categories increases. It is difficult to recognize an unseen category that is similar to the seen category. What is more, the method cannot solve cases where multiple unseen categories need to be identified at the same time. Therefore, we will commit to solving these problems in future research.

## Figures and Tables

**Figure 1 sensors-23-01668-f001:**
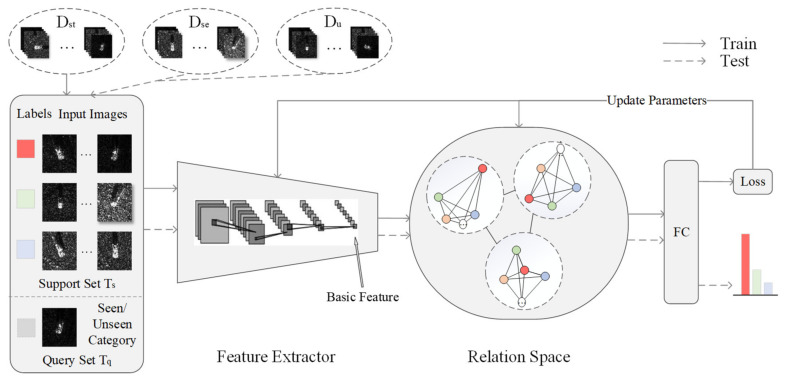
The model architecture.

**Figure 2 sensors-23-01668-f002:**
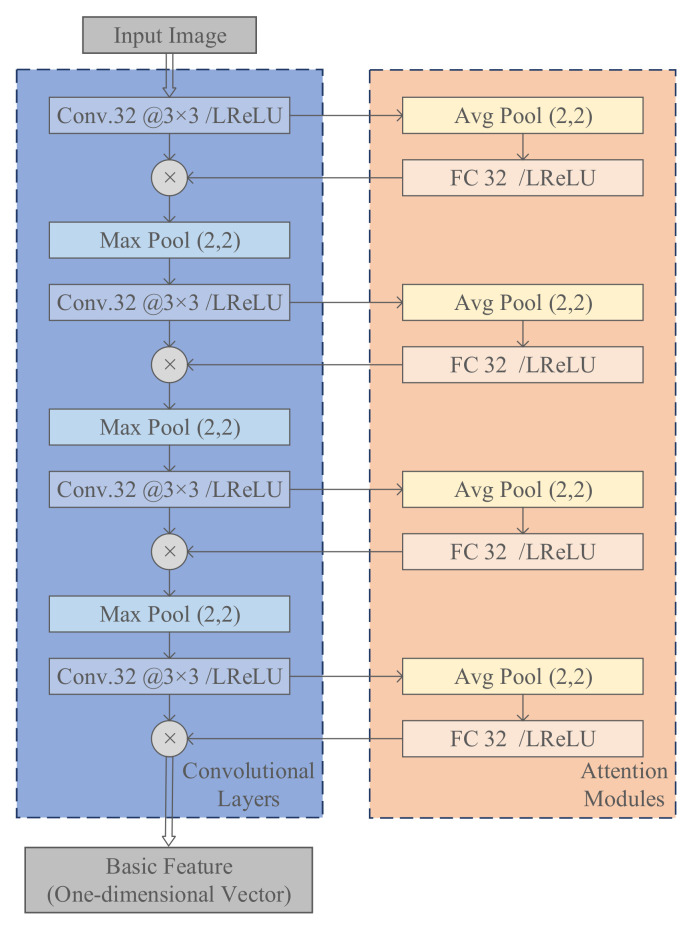
The network structure of the feature extractor.

**Figure 3 sensors-23-01668-f003:**
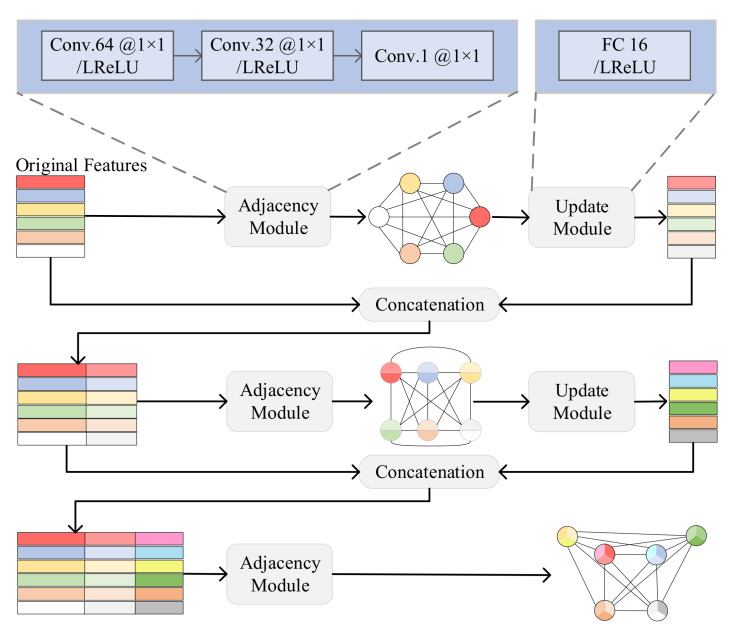
The network structure of the relation space.

**Figure 4 sensors-23-01668-f004:**
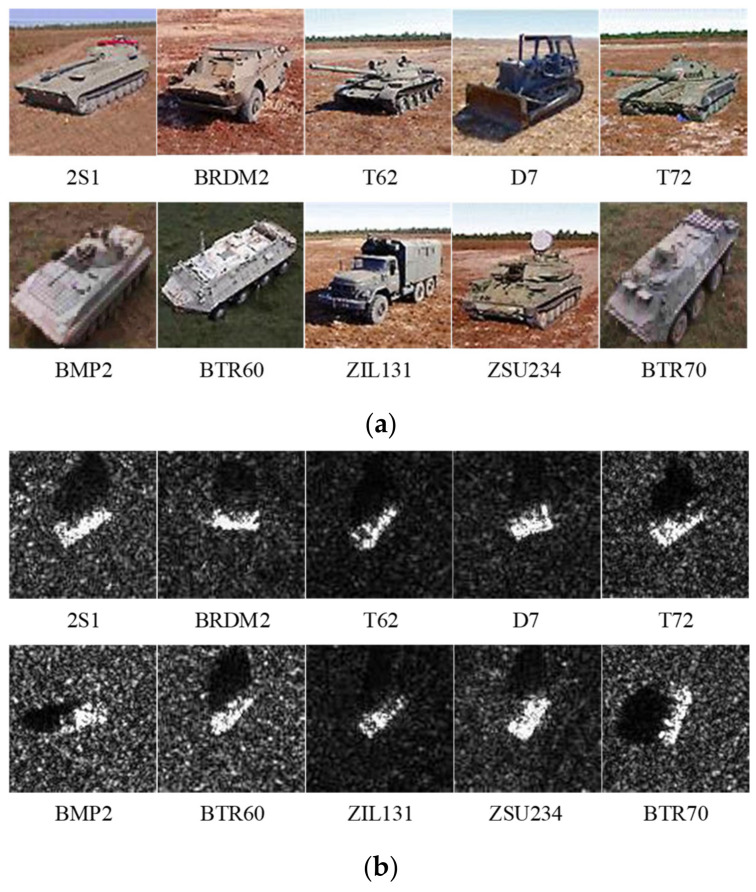
Images of each category in the MSTAR dataset. (**a**) Optical images. (**b**) SAR images.

**Figure 5 sensors-23-01668-f005:**
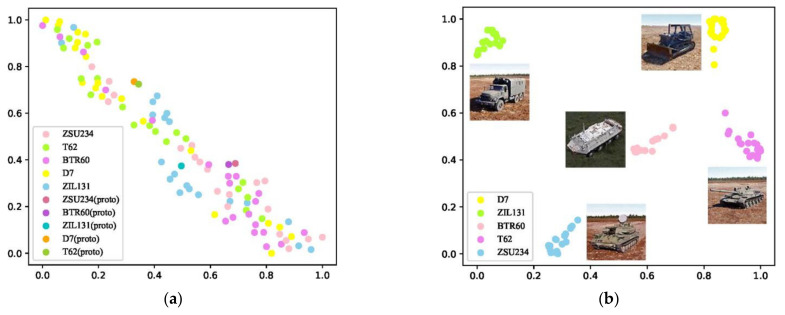
Visualization of clustering effect of (**a**) the prototype network and (**b**) the proposed model.

**Figure 6 sensors-23-01668-f006:**
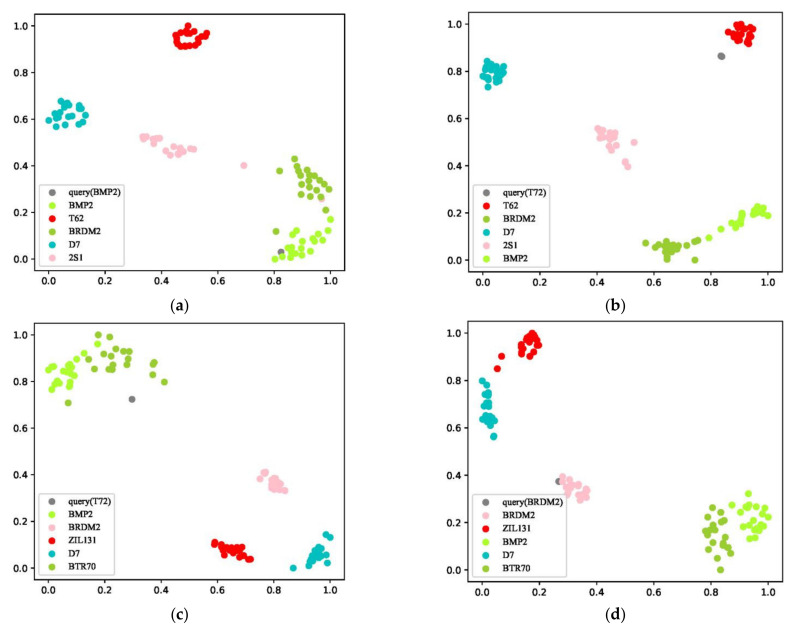
Distributions of the features of samples in different test tasks. The categories in $T_{s}$ of (**a**,**b**) are the same; the categories in $T_{s}$ of (**c**,**d**) are the same; the categories in $T_{q}$ of (**b**,**c**) are the same; the categories of (**c**,**e**) are the same, but the samples are different; the categories of (**d**,**f**) are the same, but the samples are different.

**Figure 7 sensors-23-01668-f007:**
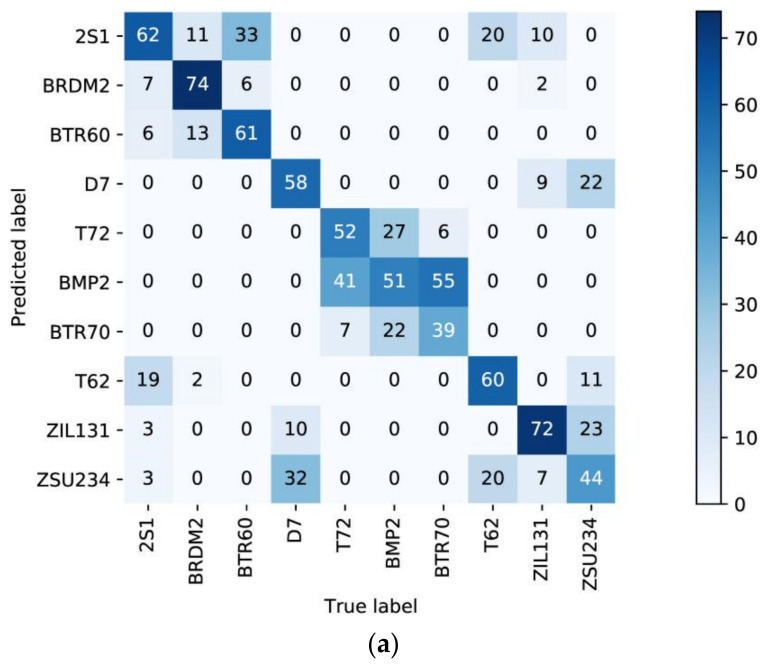
Frequencies heat map of (**a**) the proposed method, (**b**) Fea-DA, and (**c**) GAN_OSR. The same true label as the predicted label means that the class is recognized as unseen.

**Table 1 sensors-23-01668-t001:** Selection mode of T_s_ and T_q_ in training and testing phases (where are they selected from).

		$D_{u}$	$D_{s}$	
			$D_{\mathrm{st}}$	$D_{\mathrm{se}}$
Training	$T_{s}$		√	
	$T_{q}$		√	
Testing	$T_{s}$			√
	$T_{q}$	√		√

**Table 2 sensors-23-01668-t002:** Recognition accuracies for unseen categories and seen categories with different methods in the five-types classification problem (K = 20).

Method	Unseen Categories	Seen Categories
Prototype Network [3]	16.72%	16.80%
Fea-DA [19]	32.87%	78.19%
GAN_OSR [18]	45.43%	90.31%
O_SAR [16]	60.24%	42.72%
Proposed Method	52.76%	95.00%

**Table 3 sensors-23-01668-t003:** Recognition accuracies for unseen categories and seen categories under different setting conditions.

		C = 3	C = 5	C = 7
Unseen Categories	K = 10	62.89%	46.24%	31.78%
	K = 20	67.00%	52.76%	34.12%
Seen Categories	K = 10	96.92%	96.48%	93.37%
	K = 20	99.06%	96.00%	94.27%

**Table 4 sensors-23-01668-t004:** Recognition accuracies under the conditions of different unseen category proportion during training.

Ratio ^1^		3:1	1:1	1:3
Unseen Categories	C = 3	62.64%	68.50%	70.00%
	C = 5	37.30%	52.76%	57.22%
All Categories	C = 3	91.16%	92.06%	91.50%
	C = 5	88.96%	90.80%	89.05%

^1^ The ratio of the seen category and unseen category.

## Data Availability

Not applicable.

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
