# Peer review of "SAR Target Recognition with Limited Training Samples in Open Set Conditions"

_sensors, 2023, doi:10.3390/s23031668_

Round 1

Reviewer 1 Report

1)      Section 3.2 is written unclear. How rij in (6) is calculated? What is V in (7)? What is role of FC in the update module?

2)      Explanation of training and testing phase in 3.3 is inaccurate and unclear. You said that the seen data is divided into Dst and Dse. But, in another place, you call Dst as unssen category. What means query sample in the training phase? How obtain y_hat (possibility distribution of x) in (9)?

3)      The proposed method should be compared with more related works in Table 2.

4)      It is suggested that in addition to overall accuracy, the measures such as F1 score and AUC are used for comparison.

5)      In Fig. 7, the confusion matrices of all competitors should be added.

6)      The related works, which are recently published in 2022, are not cited in the paper.

Reviewer 2 Report

This paper was about the target detection from SAR images using GNN method.

The methodology had an interesting structure and the results were acceptable and well presented.

I have only some minor comments:

1. Abstract should be revised. It is better to include the numerical results in the abstract.

2. The innovation of the article is not very prominent. Please make it clear in the introduction.

3. Please make the objectives of the paper clear in the introduction.

4. The conclusion was a little weak. Please add some numerical and comparison analysis.

Reviewer 3 Report

In this paper, a task-oriented OSR method for SAR is proposed. To handle the problem of SAR OSR with limited training samples, this paper introduces graph convolutional network to identify the seen and unseen categories. Specifically, the reviewer has the following comments to this work:

1. In line 249, “T72 (tank) and BMP2 are respectively taken as the Tq” should be “BMP2 and T72 (tank)”.

2. In line 254, the category of query in (d) should be BRDM2 not BMP2 .

3. The explanation for case ratio improving the the accuracy of unseen category recognition will be welcome to be listed in the paper.

4. The authors tried to conclude the recent advances. However, there are still some relevant works about classification with limited training samples. It is recommended to add these works to the reference, such as

https://www.doi.org/10.1109/TIP.2021.3104179

https://www.doi.org/10.1109/TVT.2022.3196103

https://www.doi.org/10.3390/rs11111374

Round 2

Reviewer 3 Report

The authors have revised the manuscript according to the  comments of reviewers. I have no other questions.